# Genomic Analysis of Advanced Phyllodes Tumors Using Next-Generation Sequencing and Their Chemotherapy Response: A Retrospective Study Using the C-CAT Database

**DOI:** 10.3390/medicina60111898

**Published:** 2024-11-19

**Authors:** Shuhei Suzuki, Yosuke Saito

**Affiliations:** 1Yamagata Hereditary Tumor Research Center, Yamagata University School of Medicine, 2-2-2 Iida-nishi, Yamagata 990-9585, Japan; 2Department of Clinical Oncology, Yamagata Prefectural Shinjo Hospital, 720-1 Kanazawa, Shinjo 996-8585, Yamagata, Japan; 3Department of Gastroenterology, Yamagata City Hospital Saiseikan, 1-2-26 Nanokamachi, Yamagata 990-0042, Japan

**Keywords:** phyllodes tumor, genomic testing, chemotherapy

## Abstract

*Background and Objectives*: Phyllodes tumors are rare breast neoplasms with limited therapeutic options and poorly understood molecular characteristics. This study aimed to analyze genomic alterations and treatment outcomes in advanced phyllodes tumors using Japan’s national clinical genomic testing registry (C-CAT database) to identify potential therapeutic targets and predictive markers. *Materials and Methods*: We conducted a retrospective analysis of 60 phyllodes tumor cases from 80,329 patients registered in the C-CAT database between June 2019 and August 2024. Comprehensive genomic profiling was performed using multiple platforms including FoundationOne CDx, NCC OncoPanel, and other approved tests. Treatment responses were evaluated according to RECIST criteria, and pathogenic variants were assessed using established databases including ClinVar and OncoKB. *Results*: The cohort’s median age was 54 years (range: 13–79), with TERT promoter variants (70%), MED12 (52%), and TP53 (50%) mutations being the most frequent alterations. Forty patients received first-line chemotherapy, predominantly anthracycline-based regimens (*n* = 29). Although not reaching statistical significance, cases with CDKN2A and TERT alterations showed trends toward treatment resistance (OR > 3.0). One patient with a high tumor mutational burden (37/Mb) responded to pembrolizumab. Potential germline variants were identified in two cases (3.3%), involving MSH6 and TP53 alterations. Notably, no cases with CDKN2B alterations demonstrated treatment response (*p* = 0.09). *Conclusions*: Our findings suggest distinct molecular patterns in phyllodes tumors compared to other soft tissue sarcomas, with potential implications for treatment selection. The identification of specific genetic alterations associated with treatment resistance may guide therapeutic decision-making, while the presence of actionable mutations in select cases indicates potential opportunities for targeted therapy approaches.

## 1. Introduction

The advent of improved access to next-generation sequencing (NGS) has led to the widespread implementation of cancer genomic testing in both research and clinical settings. While the extent of genomic testing varies among different healthcare systems internationally, it has become increasingly common across multiple cancer types, including lung, thyroid, and head/neck cancers [1,2,3]. In hormone receptor-positive breast cancer, particularly in advanced or metastatic settings, the CAPItello-291 trial [4] demonstrated the efficacy of capivasertib plus fulvestrant for alterations in *PIK3CA* and *PTEN*, leading to the adoption of genomic testing platforms such as FoundationOne CDx as companion diagnostics in many countries. Furthermore, genomic testing has gained traction in rare cancers, becoming increasingly utilized in clinical practice [5]. In breast oncology, the emergence of entities such as secretory carcinoma of the breast, which shows a high detection rate of *NTRK* fusions and accessibility to TRK inhibitors, has made genomic testing an indispensable tool in breast cancer management [6].

Phyllodes tumors represent a rare neoplasm, accounting for 0.3–1.0% of breast tumors, with a notably higher prevalence among Asian populations, including Japanese populations, and Hispanic populations in Latin America [7,8,9]. While the etiology remains largely unknown, some cases have been associated with Li–Fraumeni syndrome [10]. These tumors often demonstrate resistance to chemotherapy, necessitating aggressive surgical intervention when feasible. When chemotherapy is indicated, treatment typically follows advanced soft tissue sarcoma protocols, commonly employing agents such as adriamycin, ifosfamide, or pazopanib. However, the generally poor response to chemotherapy and limited data on genetic alterations that might guide treatment decisions remain significant challenges. Recent case reports have emerged describing treatments based on genetic testing and protein expression profiles [11]. A previous comprehensive genomic profiling study of 24 phyllodes tumors revealed a high frequency of *TP53* mutations and identified potentially targetable alterations in some cases, including *KIAA1549-BRAF* fusion and *FGFR3-TACC3* fusion, emphasizing the importance of genomic testing [12].

Recent molecular studies have advanced our understanding of phyllodes tumors beyond traditional histopathological classification. The identification of recurrent mutations, particularly in *TERT* promoter regions and *MED12*, has provided new insights into tumor biology and potential therapeutic targets. These molecular findings have also proven valuable in diagnostic challenges, particularly in distinguishing phyllodes tumors from other breast neoplasms such as metaplastic carcinomas and cellular fibroadenomas [13]. The higher prevalence of phyllodes tumors in Asian populations, including Japan and other East Asian countries, has contributed to growing interest in molecular characterization studies from these regions.

In this study, we conducted a retrospective analysis of genomic data and treatment outcomes from unresectable advanced phyllodes tumors, utilizing the Center for Cancer Genomics and Advanced Therapeutics (C-CAT) database, Japan’s national clinical genomic testing registry. Given the paucity of real-world clinical data on genomic profiles and treatment responses in this rare tumor type, our findings may provide valuable insights for clinical practice. Here, we present our analysis of genomic alterations and their correlation with treatment responses in advanced phyllodes tumors.

## 2. Materials and Methods

### 2.1. Study Population

This retrospective cohort study analyzed clinical genomic testing results and clinical data from 80,329 patients registered in the C-CAT (Center for Cancer Genomics and Advanced Therapeutics) database between 1 June 2019 and August 2024. The analysis included data from multiple genomic testing platforms [14] with the following coverage periods:

NCC OncoPanel: 1 June 2019–15 August 2024; FoundationOne CDx: 1 June 2019–19 August 2024; FoundationOne Liquid CDx: 1 August 2021–17 August 2024; Guardant360 CDx: 24 July 2023–16 August 2024; GenMineTOP: 1 August 2023–16 August 2024.

The database version used for this analysis was 20240820. Patient registration was exclusively conducted through Japanese medical institutions designated by the Ministry of Health, Labour and Welfare as Core Hospitals for Cancer Genomic Medicine, Designated Hospitals for Cancer Genomic Medicine, or Collaborative Hospitals for Cancer Genomic Medicine. Only cases with written informed consent for secondary use of clinical genomic data from their respective institutions were included in the analysis.

This study was conducted with approval from both the Institutional Review Board of Yamagata University Hospital (approval number: 2023-105) and the C-CAT Database Utilization Review Board (approval number: CDU2023-032N).

### 2.2. Data Collection and Response Analysis

In this retrospective cohort study, we collected comprehensive data including demographic information (age and sex), pathological characteristics (registered histological type and tumor content), and clinical sample information specifying whether the sample was from primary or metastatic sites. We also collected details of pharmacological treatments and their responses, along with lifestyle factors including smoking and alcohol history. Family history of malignancy up to third-degree relatives was also recorded. In accordance with Japanese insurance requirements, genomic testing results were evaluated by Expert Panels comprising multiple specialists: two or more medical oncologists, pathologists (recently modified to require only one pathologist), clinical geneticists, bioinformaticians, and the treating physician or designated representative. This standardized review process ensures comprehensive evaluation of genomic findings and their clinical implications, particularly for identifying potential therapeutic targets and assessing hereditary cancer risks. Additionally, genomic characteristics were collected, including the number of gene calls, tumor mutational burden, microsatellite instability status, and specific genetic alterations identified. The pathogenicity assessment of alterations was conducted using multiple databases including ClinVar and OncoKB, with additional evaluation through the Japanese Multi Omics Reference Panel (jMorp). Variants were considered pathogenic when classified as “Likely Oncogenic” or higher in OncoKB, or when meeting C-CAT evidence level F or higher. For variant calling, a minimum allele frequency threshold of 5% was applied for tissue samples, with quality metrics including adequate coverage depth and mapping quality. Only variants classified as pathogenic were incorporated into our analysis.

Treatment response was evaluated based on first-line chemotherapeutic treatment outcomes as registered in the database. Physicians at designated institutions are required to classify responses according to RECIST criteria [15] into one of the following categories: complete response, partial response, stable disease, progressive disease, or not evaluated.

While artificial intelligence tools were not used for the primary data analysis and interpretation, AI-based language models (DeepL, Grammarly, Google Translation, ChatGPT, and Claude) were used for English language editing of the manuscript.

### 2.3. Statistical Analysis

Exploratory statistical analyses were performed using IBM SPSS Statistics for Windows, Version 25.0 (IBM Corp., Armonk, NY, USA) and Microsoft Excel 2021. This exploratory analysis focused on identifying potential associations between genetic alterations and treatment responses. Categorical variables were analyzed using chi-square tests or Fisher’s exact test when appropriate. For the analysis of treatment resistance, we defined treatment response as partial response according to RECIST criteria, while stable disease and progressive disease were classified as non-response. All statistical tests were two-sided, with findings considered exploratory in nature given the limited sample size of this rare tumor cohort. Missing data were excluded from the analysis without imputation, considering the retrospective nature of the study.

## 3. Results

### 3.1. Patients Background

The analysis cohort (Table 1) comprised 80,329 patients, including 40,447 males, 39,877 females, and 5 patients with unspecified gender. Age distribution analysis revealed that the 70–79 year age group was the largest with 23,252 patients, followed by the 60–69 year age group with 22,630 patients, and the 50–59 year age group with 16,846 patients. Among the registered alterations, TP53 was the most frequently observed with 48,176 cases, followed by KRAS with 22,344 cases, and APC with 16,446 cases. According to OncoTree classification, gastrointestinal tumors were the most common with 13,376 cases, followed by pancreatic tumors with 12,207 cases, and biliary tract tumors with 7010 cases. Breast tumors ranked fourth with 5229 cases. Within the breast tumor category, phyllodes tumors, a rare entity, accounted for 60 cases (Table 2), representing 0.07% of the total cohort and 1.1% of all breast tumors. The median age was 54 years (range: 13–79 years), and all subjects were female. Seven patients had a history of smoking, and four patients reported a history of heavy alcohol consumption. Regarding Eastern Cooperative Oncology Group Performance Status (ECOG PS), 41 patients were classified as PS 0, 14 as PS 1, 2 as PS 2, and 1 as PS 3. Family history of malignancy was present in 25 cases.

Specimens submitted for cancer genome profiling were obtained from primary tumors in 35 cases, metastatic lesions in 22 cases, blood samples in 3 cases, and unknown sources in 2 cases. Metastases were identified in 57 patients, with pulmonary metastases being most frequent (*n* = 37), followed by bone metastases (*n* = 10) and brain metastases (*n* = 3). Additional metastatic sites were documented in 26 cases, including muscle and pleural involvement.

First-line chemotherapy regimens were documented for 40 patients. Anthracycline-based regimens (including adriamycin + ifosfamide, adriamycin + cyclophosphamide, or single-agent adriamycin) were most commonly administered (*n* = 29), followed by eribulin (*n* = 4). Other agents, including docetaxel and pazopanib, were administered in seven cases.

In accordance with Japanese insurance reimbursement requirements, expert panel reviews of genomic profiling results led to new treatment recommendations in 13 cases. Among these, one patient received pembrolizumab based on high tumor mutational burden and achieved a documented response. However, in the remaining 12 cases, the recommended treatments were not administered due to geographical constraints or patient mortality. No gene-based treatment recommendations were made for 23 cases, and expert panel discussion results were pending for 14 cases.

Microsatellite instability testing was performed in 56 cases (noting that GenMine TOP and earlier versions of the NCC Oncopanel System did not include microsatellite instability testing capabilities; Table 3). Of these, 55 cases were classified as microsatellite stable, and 1 case was deemed indeterminate (Table 4).

### 3.2. Overview of Genetic Alterations and Treatment Response

We analyzed the relationship between gene alterations identified through cancer genome profiling tests and response to first-line treatment. We created a comprehensive map illustrating the associations between gene alterations in phyllodes tumors, metastatic sites, treatment responses, tumor mutational burden (/Mb), and patient age based on database findings (Figure 1).

TERT alterations were most frequently observed, with promoter variants detected in 42 cases (70%), though this frequency may be influenced by the test panel composition. This was followed by TP53 (30 cases, 50%), MED12 (29 cases, 52%), CDKN2A (24 cases, 40%), CDKN2B (15 cases, 25%), NF1 (15 cases, 25%), PTEN (15 cases, 25%), and RB1 (10 cases, 17%). It should be noted that TERT and MED12 are not included in the NCC Oncopanel System, necessitating careful interpretation of these results.

Several genes were detected in multiple cases (≤6 cases each), including MTAP, PIK3CA, EGFR, BCOR, CREBBP, KRAS, MYC, CARD11, RARA, ASXL1, DNMT3A, and KMD6A. However, in Japan, expert panels frequently discuss the possibility that ASXL1 and DNMT3A alterations may represent clonal hematopoiesis rather than somatic mutations caused by cancer.

Tumor mutational burden was reported as >0/Mb in 52 cases, with a median of 4/Mb and a maximum value of 37/Mb. The case with 37/Mb showed a positive response to Pembrolizumab treatment.

For reference, in the soft tissue sarcoma category of the C-CAT database (excluding gastrointestinal stromal tumor cases), the most frequently registered pathogenic genes were, in descending order, the following: TP53, MDM2, CDK4, and CDKN2A. While some genes, such as TP53, were commonly observed in both datasets, notable differences were identified, particularly the absence of TERT and MED12 alterations in the C-CAT database.

### 3.3. Analysis of Gene Alterations: Correlation with Treatment Response and Potential Germline Variants

Based on the results presented in Table 1, we conducted an exploratory analysis. Using chi-square tests, we examined the relationship between gene alterations and treatment resistance in 23 patients who received first-line chemotherapy with documented treatment responses. Treatment resistance was defined as cases showing no response (stable disease or progressive disease), while treatment response was defined as cases achieving partial response.

Analysis of the association between specific gene alterations and treatment outcomes (Table 3) revealed that although not reaching statistical significance, cases with CDKN2A and TERT alterations showed a trend toward treatment resistance with odds ratios exceeding 3.0. Notably, none of the cases with CDKN2B alterations demonstrated treatment response (*p* = 0.09).

Regarding hereditary tumor predisposition, our expert panel review, conducted within Japan’s insurance reimbursement framework, utilized multiple reference sources including the Ministry of Health, Labour and Welfare research findings (Kosugi Group Report) and the European Society for Medical Oncology guidelines [16]. Through this standardized expert panel review process, we identified secondary findings suggestive of hereditary cancer syndromes in two cases (3.3%), involving MSH6 and TP53 alterations, respectively.

## 4. Discussion

The implementation of genomic profiling in clinical practice has provided valuable insights into the molecular landscape of rare tumors, including phyllodes tumors of the breast. Our analysis of 60 cases from the C-CAT database represents one of the largest cohorts of phyllodes tumors with comprehensive genomic profiling to date. The findings reveal several important aspects regarding the molecular characteristics and potential therapeutic implications for this rare neoplasm.

The most frequently observed genetic alterations in our cohort were *TERT* promoter variants (70%), followed by *MED12* (52%) and *TP53* (50%) mutations. This genetic profile aligns with previous studies that have highlighted the significance of these alterations in phyllodes tumors [17,18]. Notably, our clinical experience has demonstrated the utility of genomic testing in distinguishing between metaplastic carcinoma and phyllodes tumors in challenging cases. Similar diagnostic challenges have been previously reported in the literature, with genomic profiling proving instrumental in reaching definitive diagnoses [19]. Emerging evidence suggests that DNA methylation patterns can serve as valuable diagnostic markers in differentiating these lesions [13], adding another molecular tool to the diagnostic arsenal. Furthermore, while the differentiation between phyllodes tumors and fibroadenomas often presents clinical challenges, *TERT* promoter mutations have been established as particularly useful diagnostic markers [20,21,22]. The high frequency of *TERT* promoter mutations observed in our cohort further supports their significance as a crucial molecular alteration in clinical practice.

The analysis of treatment response patterns revealed interesting correlations with specific genetic alterations. Although not reaching statistical significance, cases harboring *CDKN2A* and *TERT* alterations showed a trend toward treatment resistance (Odds Ratio > 3.0). Particularly noteworthy was the complete lack of treatment response in cases with *CDKN2B* alterations (*p* = 0.09). These findings suggest potential molecular markers that might predict treatment outcomes, though further validation in larger cohorts is necessary.

In our previous analysis of soft tissue sarcoma cases registered in the C-CAT database, we observed distinct patterns in terms of both molecular alterations and treatment responses. While soft tissue sarcomas showed a relatively low frequency of *MDM2* and *CDK4* amplifications, they demonstrated a preliminary trend where wild-type *TP53* status correlated with better treatment outcomes. Interestingly, phyllodes tumors, despite being classified as fibroblastic neoplasms, did not show this same correlation. This disparity suggests potential differences in molecular pathogenesis and treatment response mechanisms between phyllodes tumors and other soft tissue sarcomas, despite their classification within the broader category of fibroblastic tumors. The analysis of treatment response patterns by genomic alterations provides potential insights for clinical decision-making. While statistical significance was not reached due to cohort size limitations, the observed trends suggest possible molecular markers for treatment response prediction. Particularly noteworthy was the complete lack of treatment response in cases with *CDKN2B* alterations, and the trend toward treatment resistance in cases with *CDKN2A* and *TERT* alterations (OR > 3.0). These findings, although preliminary, may help inform future therapeutic strategies. Moreover, our observation of a response to pembrolizumab in a case with high tumor mutational burden suggests the potential utility of comprehensive genomic profiling in identifying candidates for immunotherapy approaches, even in traditionally chemotherapy-resistant cases.

The identification of microsatellite stability status in our cohort, with 55 out of 56 tested cases being microsatellite stable, provides important information regarding potential immunotherapy applications. However, the observation of one case with high tumor mutational burden responding to pembrolizumab suggests that genomic profiling might identify subset of patients who could benefit from immunotherapy approaches [23].

The detection of potential germline variants in 3.3% of cases (involving *MSH6* and *TP53*) highlights the importance of considering hereditary cancer syndromes in patients with phyllodes tumors. This finding aligns with previous reports of phyllodes tumors occurring in the context of Li–Fraumeni syndrome and other hereditary cancer predisposition syndromes [24].

Our study has several limitations. The retrospective nature and the potential selection bias inherent in a registry-based study should be considered when interpreting the results. Additionally, the variation in genomic testing platforms used, particularly regarding the detection of specific alterations such as *TERT* promoter variants and *MED12* mutations, which were not included in all panels, may affect the reported frequencies of these alterations. The accuracy of data entry and classification in large-scale registry databases also warrants careful consideration, as these factors could potentially impact our findings.

These findings suggest several directions for future investigation. First, the trends observed in treatment resistance patterns warrant validation in larger, preferably prospective cohorts. International collaboration would be particularly valuable given the rarity of this tumor type. Second, the role of *CDKN2A/B* alterations in treatment resistance mechanisms requires further molecular characterization. Finally, the potential of immunotherapy in genomically selected cases, particularly those with high tumor mutational burden, deserves prospective evaluation. Such studies could help establish more precise, molecularly guided treatment approaches for this challenging disease.

## 5. Conclusions

This comprehensive genomic profiling study of phyllodes tumors provides insights into their molecular landscape and potential therapeutic implications, suggesting the value of genomic testing in advancing our understanding of this challenging disease.

## Figures and Tables

**Figure 1 medicina-60-01898-f001:**
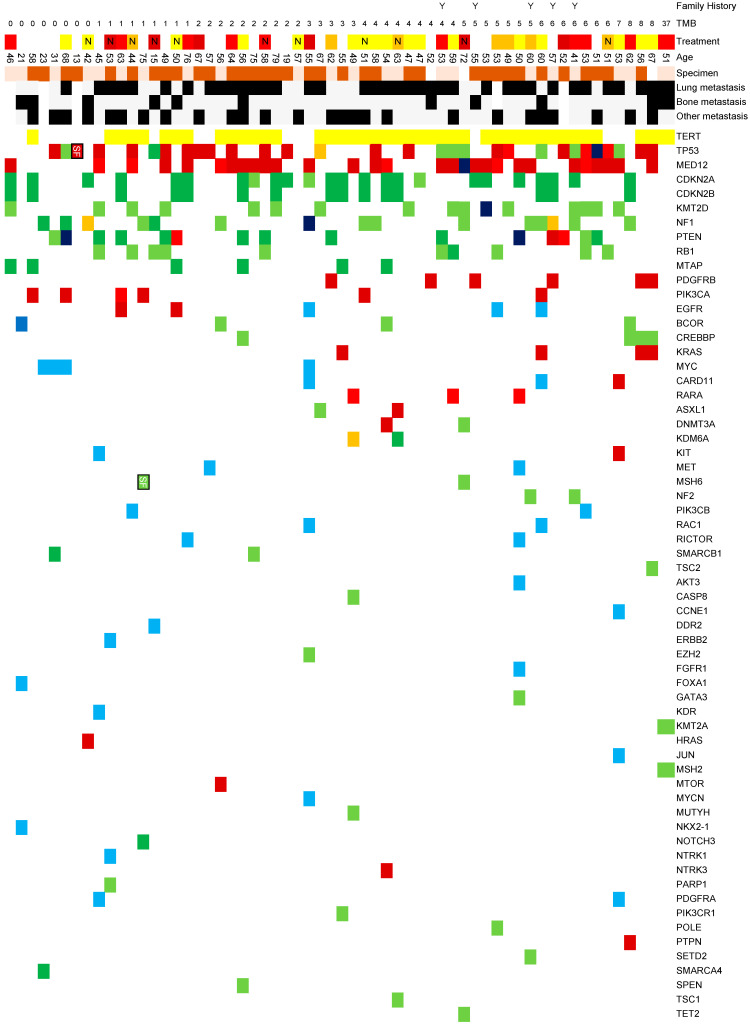
Genomic alterations and treatment responses of phyllodes tumor cases registered in the Center for Cancer Genomics and Advanced Therapeutics. Notable germline findings include MSH6 (T719fs*17, AF = 0.28) and TP53 (G245S, AF = 0.77) alterations, both classified as presumed germline pathogenic variants (PGPVs) according to Japanese PGPV criteria (Kosugi Group Criteria Ver.4; AF cutoff >0.2 for truncating variants in MSS tumors and >0.3 for TP53 variants in patients ≤30 years). TMB: Tumor Mutational Burden; PR: Partial Response; SD: Stable Disease; PD: Progressive Disease; NE: not evaluated; F: FoundationOne^®^ CDx; FL: FoundationOne^®^ Liquid CDx; N: NCC Oncopanel System; T: GenMine™ TOP Cancer Panel. For treatment regimens, unmarked cases indicate adriamycin-based therapy while “N” indicates non-adriamycin regimens (including eribulin, pazopanib, and docetaxel). Y marks indicate patients aged 40 years or younger at diagnosis.

**Table 1 medicina-60-01898-t001:** Background of cases registered in the Center for Cancer Genomics and Advanced Therapeutics.

Total Cases (*n* = 80,329)			
Primary Site		Cancer Genomics Panel	
Colorectal	13,376	FoundationOne CDx	57,085
Pancreas	12,207	FoundationOne Liquid CDx	12,184
Bile Duct	7010	NCC Oncopanel System	8016
Breast	5229	GenMine^TM^ TOP Cancer Panel	1853
Esophagus/Stomach	4937	Guardant360 CDx	1191
Prostate	4912		
Lung	4776	**Age Group (years)**	
Ovary/Fallopian Yube	4507	70–79	23,252
Soft Tissue	3197	60–69	22,630
Uterus	2775	50–59	16,846
Others	17,403	40–49	8661
		80–89	3528
		30–29	2840
**Sex**		20–29	949
Male	40,447	10–19	849
Female	39,877	0–9	718
Unknown	5	90-	56

The study period for each genomic profiling test was as follows: NCC Oncopanel System (1 June 2019 to 15 August 2024); FoundationOne^®^ CDx (1 June 2019 to 19 August 2024); FoundationOne^®^ Liquid CDx (1 August 2021 to 17 August 2024); Guardant360^®^ CDx (24 July 2023 to 16 August 2024); and GenMine^TM^ TOP Cancer Panel (1 August 2023 to 16 August 2024).

**Table 2 medicina-60-01898-t002:** Background of phyllodes tumor cases registered in the Center for Cancer Genomics and Advanced Therapeutics.

Total Phyllodes Tumor Cases (N = 60, All Female)			
Age Group (Years; Median 54)		Cancer Genomics Panel	
50–59	28	FoundationOne CDx	51
60–69	11	NCC Oncopanel System	4
40–49	10	FoundationOne Liquid CDx	3
70–79	5	GenMine^TM^ TOP Cancer Panel	2
10–19	3		
20–29	2	**ECOG PS**	
30–39	1	0	41
		1	14
**Smoking History**		2	2
No	52	3	1
Yes	7	Unknown	2
Unknown	1		
		**Metastatic sites**	
**Drinking History**		Lung	37
No	47	Bone	10
Yes	4	Brain	3
Unknown	1	Other Sites	26
		None	3
**Family History (Cancer)**		**First-Line Chemotherapy**	
Yes	31	Anthracycline-based	29
No	25	Eribulin	4
Unknown	4	Others	7

The study period for each genomic profiling test was as follows: NCC Oncopanel System (1 June 2019 to 15 August 2024); FoundationOne^®^ CDx (1 June 2019 to 19 August 2024); FoundationOne^®^ Liquid CDx (1 August 2021 to 17 August 2024); Guardant360^®^ CDx (24 July 2023 to 16 August 2024); and GenMine^TM^ TOP Cancer Panel (1 August 2023 to 16 August 2024). ECOG PS: Eastern Cooperative Oncology Group Performance Status.

**Table 3 medicina-60-01898-t003:** Comparison of cancer genomic profiling tests used in this study.

Features	FoundationOne CDx	FoundationOne Liquid CDx	NCC Oncopanel System	GenMine TOP Cancer Panel
Sample Type	FFPE tissue	Blood	FFPE tissue	FFPE tissue
Number of Genes	324	324	114	723
*TERT* Promoter Detection	Yes	Yes	No	Yes
MSI Testing	Yes	Yes	Yes *	No
TMB Assessment	Yes	Yes	Yes	Yes
Minimum Tumor Content Required	20%	N/A	20%	20%
Required DNA Input	50 ng	2 tubes	50 ng	50 ng

*: Earlier versions did not include MSI testing capability. FFPE: Formalin-Fixed Paraffin-Embedded; MSI: Microsatellite Instability; TMB: Tumor Mutational Burden; N/A: Not Applicable.

**Table 4 medicina-60-01898-t004:** Result of phyllodes tumor cancer genomic testing cases registered in the Center for Cancer Genomics and Advanced Therapeutics.

Total Phyllodes Tumor Cases (N = 60)				
Druggable Variants: Expert Panel Review		Treatment ** Resistance-Associated Gene Alterations		
No	23	Gene	OR	*p*-value
Yes, Treatment NOT Administered	12	CDKN2A	3.5	0.28
Yes, Treatment Administered	1 *	CDKN2B	INF	0.09
Unknown	14	KMT2D ***	1.7	0.68
		MED12 ***	2.8	0.32
**Microsatellite Instability**		RB1	2.7	0.39
Microsatellite Stable	52	PTEN	2.1	0.54
No Result	1	TERT ***	3.1	0.29
Not Included	5	TP53	0.8	0.86
**Presumed Germline Pathogenic Variant**				
No	58			
Yes	2			

*: High Pembrorizumab to Tumor Mutation Burden; **: First-Line Chemotherapy; ***: Not Included in NCC Oncopanel System; INF: Infinity (no clinical response was observed in cases with CDKN2B alterations).

## Data Availability

The dataset generated during this study is not publicly accessible due to confidentiality agreements as part of the ethics approval process.

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
