# Peer review of "Genomic Analysis of Advanced Phyllodes Tumors Using Next-Generation Sequencing and Their Chemotherapy Response: A Retrospective Study Using the C-CAT Database"

_medicina, 2024, doi:10.3390/medicina60111898_

Round 1

Reviewer 1 Report

Comments and Suggestions for Authors

·         Brief summary: This study by Suzuki and Saito analyzed genomic alterations in 60 phyllodes tumor cases from Japan's C-CAT database, identifying frequent mutations in TERT, MED12, and TP53. Trends indicated possible treatment resistance linked to CDKN2A and TERT mutations, while one high-mutation-burden case responded to pembrolizumab. Their findings highlight unique molecular patterns in phyllodes tumors, suggesting opportunities for targeted therapies.

ü  Specific comments:

o    Although relatively large for rare type of phyllodes tumors is studied by the authors, the sample size of 60 patients may still be too small to reach statistical significance for some findings, particularly in assessing treatment response.

o    There is lack of functional studies which are needed to confirm causative links and fully understand the mechanisms behind resistance or responsiveness.

o    Since the data is specific to a Japanese population, findings may not be entirely generalizable to other ethnic groups or regions.

o    Overall, the study is a valuable step forward in understanding the molecular landscape of rare cancer type such as phyllodes tumors, though follow-up studies with larger, more diverse cohorts and prospective designs could strengthen and expand on these insights.

Author Response

Response to Reviewer #1:

We sincerely thank the reviewer for their thorough evaluation and accurate summary of our work. We particularly appreciate the reviewer's recognition of our study's contribution to understanding the molecular landscape of phyllodes tumors.

1. Regarding sample size:
Reviewer's comment: "Although relatively large for rare type of phyllodes tumors is studied by the authors, the sample size of 60 patients may still be too small to reach statistical significance for some findings, particularly in assessing treatment response."

Our response: We fully agree with this important point. While our cohort represents one of the largest genomic studies of phyllodes tumors to date, we acknowledge the statistical limitations. We have expanded our discussion of this limitation and clearly stated the exploratory nature of our statistical analyses in the Methods section. We have also added more detailed statistical methodology information to ensure transparent reporting of our findings.

2. Regarding functional studies:
Reviewer's comment: "There is lack of functional studies which are needed to confirm causative links and fully understand the mechanisms behind resistance or responsiveness."

Our response: We agree that functional validation would strengthen our findings. We have expanded our discussion to include this limitation and have outlined specific directions for future functional studies in the Discussion section, particularly regarding the role of CDKN2A/B alterations in treatment resistance mechanisms.

3. Regarding population specificity:
Reviewer's comment: "Since the data is specific to a Japanese population, findings may not be entirely generalizable to other ethnic groups or regions."

Our response: We acknowledge this limitation and have expanded our discussion of the population characteristics. We have also added more context regarding the known higher prevalence of phyllodes tumors in Asian populations, while noting the importance of international validation studies in our future directions.

4. Regarding follow-up studies:
Reviewer's comment: "Overall, the study is a valuable step forward in understanding the molecular landscape of rare cancer type such as phyllodes tumors, though follow-up studies with larger, more diverse cohorts and prospective designs could strengthen and expand on these insights."

Our response: We appreciate this constructive suggestion and have added a new paragraph in the Discussion section outlining specific directions for future research, including the need for international collaborative efforts and prospective validation studies.

Specific changes made in response to these comments include:
1. Enhanced statistical analysis description in Methods section
2. Added comparative analysis of genomic testing platforms
3. Expanded Discussion section with future research directions
4. Added limitations discussion regarding cohort characteristics

Reviewer 2 Report

Comments and Suggestions for Authors

Phylloid tumors of the breast represent a relatively rare fibroepithelial tumor that exhibits a morphologic continuum, ranging from benign to malignant. The majority of phylloid tumors behave in a benign manner, with local recurrence occurring in a relatively small proportion of cases. In rare instances, the tumor may metastasize. The manuscript, entitled "Genomic Analysis of Advanced Phyllodes Tumors Using Next-Generation Sequencing and Their Chemotherapy Response," The study, entitled "A Retrospective Study Using the C-CAT Database," is a comprehensive analysis of the full-genome sequencing of this tumor type and a detailed comparison of the molecular profile of the tumor and the response to treatment. The article demonstrates a high level of competence in its use of the most modern molecular genetic methods. The reviewer has provided a few minor comments.

1. It would be beneficial to provide a more detailed characterization of the sample and to include more comprehensive data regarding the medical history of the patients. In particular, the hereditary component requires further investigation. Furthermore, a separate table should be created to organize these data.

2. If the authors utilized predictor programs, please provide a detailed description of the algorithms employed to detect the pathogenicity of the identified mutations.

3. It is essential to provide a comprehensive overview of the spectrum of identified mutations, with an emphasis on incorporating gene-phenotypic correlations wherever possible.

The aforementioned observations do not detract from the intrinsic value of the research conducted. The work is of sufficient quality.

Author Response

We deeply appreciate the reviewer's thorough evaluation and constructive comments on our manuscript. We are particularly grateful for the reviewer's recognition of our study's contribution to understanding this rare tumor type.

1. Regarding patient medical history:
Reviewer's comment: "It would be beneficial to provide a more detailed characterization of the sample and to include more comprehensive data regarding the medical history of the patients. In particular, the hereditary component requires further investigation. Furthermore, a separate table should be created to organize these data."

Our response: We thank the reviewer for this valuable suggestion. We have added more detailed information about the hereditary component, specifically:
- Added detailed information about the presumed germline pathogenic variants in Figure 1's legend, including specific mutations (MSH6: T719fs*17, AF=0.28; TP53: G245S, AF=0.77) and their classification criteria according to Japanese PGPV guidelines
- Expanded the description of the Expert Panel evaluation process regarding hereditary cancer assessment in the Methods section
- Enhanced the discussion of hereditary cancer implications in the Discussion section

2. Regarding predictor programs:
Reviewer's comment: "If the authors utilized predictor programs, please provide a detailed description of the algorithms employed to detect the pathogenicity of the identified mutations."

Our response: We have expanded our Methods section to include more detailed information about variant assessment criteria:
- Added specific criteria for variant classification (OncoKB 'Likely Oncogenic' or higher, C-CAT evidence level F or higher)
- Included quality metrics such as allele frequency thresholds and coverage requirements
- Added Table 3 comparing the technical specifications and capabilities of different genomic profiling platforms used in this study

3. Regarding mutation spectrum:
Reviewer's comment: "It is essential to provide a comprehensive overview of the spectrum of identified mutations, with an emphasis on incorporating gene-phenotypic correlations wherever possible."

Our response: We have enhanced our presentation of mutation data by:
- Expanding the Results section with more detailed analysis of mutation patterns
- Adding genotype-phenotype correlations, particularly regarding treatment responses
- Including detailed analysis of age-related patterns (marked with 'Y' in Figure 1 for patients ≤40 years)

We believe these revisions have significantly improved the comprehensiveness and clarity of our manuscript while maintaining its scientific rigor. We are grateful for the reviewer's insightful comments that helped us achieve this improvement.